# Polyphosphazene-Based Biomaterials for Biomedical Applications

**DOI:** 10.3390/ijms232415993

**Published:** 2022-12-15

**Authors:** Geun-Woo Jin, N. Sanoj Rejinold, Jin-Ho Choy

**Affiliations:** 1Intelligent Nanohybrid Materials Laboratory (INML), Institute of Tissue Regeneration Engineering (ITREN), Dankook University, Cheonan 31116, Republic of Korea; 2R&D Center, CnPharm Co., Ltd., Seoul 03759, Republic of Korea; 3Department of Pre-Medical Course, College of Medicine, Dankook University, Cheonan 31116, Republic of Korea; 4International Research Frontier Initiative (IRFI), Tokyo Institute of Technology, Yokohama 226-8503, Japan; 5Institute of Innovative Research, Tokyo Institute of Technology, Yokohama 226-8503, Japan

**Keywords:** polyphosphazenes (PPZs), drug delivery, biodegradability, biocompatibility

## Abstract

Recently, synthetic polymers have attracted great interest in the field of biomedical science. Among these, polyphosphazenes (PPZs) are regarded as one of the most promising materials, due to their structural flexibility and biodegradability compared to other materials. PPZs have been developed through numerous studies. In particular, multi-functionalized PPZs have been proven to be potential biomaterials in various forms, such as nanoparticles (NPs) and hydrogels, through the introduction of various functional groups. Thus, PPZs have been applied for the delivery of therapeutic molecules (low molecular weight drugs, genes and proteins), bioimaging, phototherapy, bone regeneration, dental liners, modifiers and medical devices. The main goal of the present review is to highlight the recent and the most notable existing PPZ-based biomaterials for aforementioned applications, with future perspectives in mind.

## 1. Introduction

As the field of clinical research develops and the need for advanced biomaterials increases, the need for new materials also increases. Compared to natural polymers, synthetic polymers are regarded as better materials, because they are well-defined and easy to modulate for multipurpose medical applications [1,2,3,4,5]. Among these, polyphosphazenes (PPZs) are a new type of polymers, distinct from existing synthetic polymers due to their synthetic flexibility and bio-degradability [6,7]. PPZs are synthetic polymers with an inorganic backbone composed of repeating units of phosphorous and nitrogen atoms. The two electrons of the phosphorus can be used for side-chain conjugation, while two electrons of nitrogen remain a lone pair. Compared to other synthetic polymers, such as polylactic acid (PLA) and poly(lactide-*co*-glycolide) PLGA, the advantages of PPZ are remarkable, due to the controlled tuning of physico-chemical properties via the side-chain conjugation. Furthermore, PPZs are hydrolytic and biodegradable materials yielding nontoxic degradation products. Such advantages make PPZs a versatile biomaterial to be used in a wide range of biomedical applications, such as drug delivery, bioimaging, phototherapy, bone regeneration, dentistry (restorative composites and dental liners) and medical devices. In the present review, therefore, we focus on the latest developments and prospects of PPZs in the aforementioned biomedical applications.

## 2. History of Polyphosphazenes Synthesis

In 1895, the first synthesis of PPZs was reported by H.N. Stokes via thermal ring-opening polymerization of hexachlorocyclotriphosphazene (HCCP) [8]. However, the first PPZs was a cross-linked rubbery material described as ‘inorganic rubber’ that is insoluble in all solvents [9]. In 1965, Allcock and Kugel synthesized the first stable PPZs via thermal ring-opening polymerization of HCCP [10]. They succeeded in synthesizing poly(organophosphazene)s by substituting chlorine atoms on linear PPZs with alkoxy- and aryloxyphosphonitriles (Figure 1a). The substitution reaction was possible as the PPZs were soluble in organic solvents, such as benzene. Although these researchers opened the way to the practical application of PPZs, the molecular weight distribution was too broad. The substitution reaction can also be achieved by nucleophilic substitution reaction with HCCP and nucleophiles to produce cyclotriphosphazene. Further multiple substitution reactions lead to a net-like structure. The cross-linked cyclotriphosphazene have various morphologies, including nanofibers, nanotubes and nanosheets [11] (Figure 1a).

The synthesis of PPZs with narrow molecular weight distribution was achieved by living cationic polymerization at ambient temperature by Allcock et al. in 1996 (Figure 1b) [12]. This living cationic polymerization method contributed not only to control the molecular weight, but also to develop a wide variety of PPZs. The direct synthesis of poly(organophosphazene) was enabled by living cationic polymerization using organo-substituted phosphoranimines, organophosphoranimines (Figure 1c) [13]. The living chain ends of phosphoranimine also provide a material basis for the synthesis of block PPZs. Because of the presence of living sites at the ends of polymer chain in living cationic polymerization, further chemical reactions are possible with other phosphazene monomers, to make block copolymers [14].

## 3. Advantages of Polyphosphazenes in Biomedical Application

Based on the established synthetic method of PPZs by Allcock and colleagues, extensive applications of PPZs have been attempted. PPZs, conjugated with various side groups around a degradable backbone, have been studied. Such studies have provided evidence for the biological applicability of PPZs as promising materials for application in the biomedical field. Some of the very recently reported biomedical applications of PPZ-based biomaterials are shown in Table 1.

The following focuses on the advantages of PPZs in biomedical applications, including structural/functional diversity, biodegradability and biocompatibility. 

### 3.1. Structural/Functional Diversity

Poly(organophosphazene) is synthesized by substituting the chlorine atoms of poly(dichlorophosphazene) with organic nucleophiles, which include alkoxides, primary or secondary amines. This synthetic method enables the preparation of biomaterials with great versatility, allowing for the modulation of physical properties, such as hydrophilicity/hydrophobicity, degradability and responsiveness to external stimuli, as below.

In aqueous media, amphiphilic PPZs are self-assembled to form micelles. The hydrophobic core can be utilized as a reservoir for hydrophobic drugs, thereby improving the drug delivery efficiency. For example, Qui et al. synthesized the amphiphilic graft PPZs with thermal ring-opening polymerization and a subsequent substitution reaction of hydrophilic polyethylene glycol (PEG) and hydrophobic ethyl tryptophan (EtTrp). DOX was physically entrapped in the polymeric micelles [15]. In another recent study, fluorescein and quercetin were co-loaded into polycondensed octachlorocyclotetraphosphazene for imaging and cancer therapy [16]. Similarly, a dual drug approach was applied for developing cyclomatrix PPZ, wherein drugs, such as curcumin and quercetin, were loaded for pH-responsive drug delivery systems for anti-cancer therapy. Despite the fact that particle size and morphology were ideal, the major problem associated with this study was the lack of in vitro and in vivo therapeutic benefits of the developed system [17].

PPZs are hydrolytic, and it was found that their degradability can be modulated by the substituents. The introduction of depsipeptide esters to PPZs accelerated the degradation, increasing hydrolysis rates [21].

Attempts have been made to synthesize external stimuli-sensitive PPZs. Thermo-sensitive PPZs containing multiple thiol (–SH) groups were synthesized by Potta et al. [22]. The aqueous solutions of these polymers were changed into hydrogel at body temperature, and the gel strength was further strengthened by the cross-linking of thiol groups. pH-sensitive hydrogel has also been prepared from PPZs functionalized with sodium oxybenzoate and methoxyethoxyethoxy side groups [23]. A dye diffusion study using the hydrogels showed a pH-sensitive drug release pattern comprising complete release in 4–12 h at pH 7.4, but lower release at pH 2, even after 48 h. 

### 3.2. Biodegradability 

The biodegradability of PPZs is beneficial in many medical applications. The hydrolysis mechanism of PPZs is still controversial; whether side groups are eliminated first or side groups accelerate PPZs backbone hydrolysis is debated [24]. However, it is generally believed that the degradation of PPZs is triggered in acidic media (Figure 2), and that the protonation of the phosphorus atom induces such an acidic-driven degradation, as represented in Figure 2 [24]. In any cases, the degradation products derived from the backbone is non-toxic and neutral ammonium phosphate. These degradation products are beneficial when compared to other biodegradable polymer-based ones, which produce acid (i.e., lactic acid) that can affect the delivered drug (genes and proteins).

As explained briefly in Section 3.1, the degradation rate of PPZs is known to be dependent on the substituting side groups. Allcock et al. synthesized various types of PPZs substituted with amino acid esters and alkoxide. Interestingly, the PPZs showed various degradation half-lives (in PBS, 37 °C), from a few days to 1 year: alkoxide (T_1/2_~7 days), ethyl glycinate (T_1/2_~3 months), alanine (T_1/2_~6 months), phenylalanine ethyl ester (T_1/2_~1 year) and valine (T_1/2_~1 year). These examples show that the degradability of PPZs can be modulated by selecting side groups appropriate for the medical application [25,26,27].

### 3.3. Biocompatibility

For medical applications, the safety of materials is a key consideration. The biocompatibility of PPZs have been evaluated for their application in tissue engineering; they have showed good biocompatibility [28,29]. It was found that the osteoblast cells grown on PPZ surfaces could retain their normal characteristics without perturbing their phenotype, differentiation, maturation and mineralization. The biocompatibility of intravenously injected PPZs has also been studied. The PPZs conjugated with ethyl-p-aminobenzoate (EAB) and N,N-diisopropylethylenediamine (DPA) showed favorable safety properties in both vitro and in vivo studies [30]. 

The biocompatibility is strongly supported by the fact that coronary stents—Cobra PzF stents (CeloNova Biosciences Inc., San Antonio, TX, USA)—that are coated with PPZs have already been approved by the FDA [31,32,33]. 

## 4. Biomedical Applications of Polyphosphazenes

Biomedical research on PPZs has been performed in various fields, including drug delivery, bioimaging, phototherapy, vaccine and bone regenerative engineering. The intriguing characteristic of biomaterials in a wide range of medical application is the limited diversity. The substitution reaction allows the incorporation of several functional groups into the PPZ backbone. Hence, PPZs incorporated drugs into their structure by physical encapsulation or chemical bonding, to form a prodrug delivery system. The possibility of synthesizing materials with tailored degradation kinetics and their structural flexibility makes PPZs an attractive choice for medical applications, such as a delivery carrier and for bone regeneration.

### 4.1. Small Molecule Drug Delivery

Small molecule drugs, especially cytotoxic anti-cancer drugs, have unfavorable side effects. To prevent the side effects, the release of drugs needs to be controlled using a drug delivery system. Controlled release has been achieved in the form of prodrug nanoparticles or hydrogels [34].

A PPZ-docetaxel (DTX) conjugate, named “Polytaxel”, was prepared with acid-labile *cis*-aconitic amide bonds. The PPZ-DTX was self-assembled to form stable micelles (41.8 nm), demonstrating good tumor-targeting properties and thereby excellent anti-cancer activity in a gastric tumor mouse model [35]. A PPZ-Pt conjugate was synthesized by the same group. The platinum-based anti-cancer moiety, (dach)Pt, was conjugated to a PPZ-based carrier polymer using cis-aconitic acid as a linker, resulting in a PPZ-Pt conjugate named “Polyplatin”. Polyplatin was self-assembled to form nanoparticles with a mean diameter of 55.1 nm. Polyplatin exhibited good anti-tumor activity in a gastric tumor model, with low systemic toxicity [36]. 

Generally, a combinatorial approach is preferred for treating aggressive tumors with multi-drug resistance (MDR). Although many such studies have been conducted, the practical difficulty in coordinating pharmacokinetic effects, and thus understanding their synergism, is a reality. Attempts have been reported to improve the simultaneous delivery of multiple drugs to the desired site. For example, a drug self-framed delivery system (DSFDS) has been made using multiple drugs as monomers for constructing cyclomatrix polyphosphazene nanoparticles (CPPZ NPs). Notably, it is an extremely flexible common platform to realize the rational design of combination therapy, which is verified by delivering DOX with mitoxantrone (Mit), resveratrol (RES), curcumin (Cur) and porphyrin (TPP). To execute the proof of concept, DOX-RES-CysM-CPPZ NP was tested for the therapeutic efficacy of MDR tumors. These experimental results are promising for the use of DSFDS as an efficient combination therapeutic system for cancer treatment [18].

Similarly, a CPT (camtothecin) delivery system was made of cross-linked poly(cyclotriphosphazene-co-phloretin) (PCTPPT) microspheres and was well characterized. As per the FT-IR spectra, the appearance of new bands at 1009 and 1132 cm^−1^ confirmed the successful polymerization of HCCP with phloretin (Pht). Thermogravimetric analysis (TGA) of the PCTPPT microspheres showed that the increased thermal stability could be due to their highly cross-linked, covalently bonded structure. Additionally, the particle size of the microspheres are inversely proportional to the HCCP:Pht mole ratio. These microspheres were able to induce controlled release of CPT [19].

In another study, an oral immuno-drug delivery system was constructed of a phospholipid-like amphiphilic PPZ nano-vesicle (RGD-PEOP) that structurally mimicked a bacterial-membrane (MBM). Ovalbumin (OVA) antigen was loaded into this delivery system to evoke maximum anti-tumor immune response. The MBM structure was created by phospholipid-like octadecylphosphoethanolamine groups in the vesicle membrane, to enhance OVA loading through salt bridge and hydrogen bonding interactions. Further, to impart bacterial membrane function, RGD peptide was linked to the surface of the particles, creating a 3.4-fold higher transfection efficiency than the intact OVA. Both in vitro and in vivo analyses showed that the MBM nano-vesicle could induce co-stimulatory molecules and MHC class II complexes effectively. In addition, IFN-γ and IL–4 levels were also improved after treatment with the MBM nano-vesicle group, resulting in maximum efficacy (~69%) against E.G7-OVA tumors [20].

Al-Abd et al. synthesized an injectable and thermosensitive hydrogel that transforms from solid to gel at body temperature. Anti-cancer drug DOX was formulated at 0.6% into a 10% PPZ hydrogel. According to the result, 40% and 90% of DOX was slowly released over 5 weeks in vitro and in vivo, respectively. Due to the sustained release profile, T_1/2β, tumor_ (drug half-life at tumor) was 1.8-fold longer than that of DOX, but C_max, tumor_ was 2.2-fold lower in producing a similar AUC_tumor_ and antitumor activity. However, the PPZ hydrogel system showed lower systemic toxicity [37]. 

Zhou et al. prepared a PPZ nano-drug through the reaction of HCCP with 4-hydroxy-benzoic acid (4-hydroxy-benzylidene)-hydrazide (HBHBH) and DOX (Figure 3). The nano-drug showed sustained DOX release, due to the acid-labile acylhydrazone bond in HBHBH. The nano-drug remained stable at pH 7.4, but was degraded and released DOX in an acidic environment such as tumors (pH 6.8) and lysosome and endosome (pH 5.0). In particular, the DOX contents could be tuned from 10.6% to 52.6% by changing the dosing ratio of DOX to HBHBH. The nano-drugs showed excellent toxicity to tumor cells and reduced the side effects to normal cells both in vitro and in vivo, due to their pH-sensitive properties [38].

Although DOX has been used for many types of cancer, the major associated issue is drug resistance, which can limit the clinical applicability. Xu et al. tried to overcome this problem by using the self-assembled behavior of amphiphilic PPZs. They constructed polymersomes loading DOX and chloroquine phosphate (CQ), a drug resistance-reversal agent via dialysis method. The cytotoxicity of the dual drug-loaded polymersome was performed on DOX-resistant breast cancer (MCF–7/Adr) and leukemia (HL60/Adr) cells. As a result, the polymersome containing DOX and CQ at 1:1 weight ratio exhibited the strongest toxicity against cancer cells. Furthermore, the in vivo anti-tumor efficacy study carried out on a zebrafish xenograft model also showed that the polymersome made an outstanding contribution to improve the sensitivity of breast cancer to chemotherapy. These findings suggest that this DOX and CQ co-delivery system based on PPZ polymersomes might be promising for drug resistance reversal in cancer therapy [39].

In another study, the anti-cancer drugs camptothecin (CPT) and epirubicin (EPI) were delivered using a PPZ nanocarrier synthesized by Quiñones et al. Linear PPZ was modified with lipophilic tocopherol or testosterone glycinate and with hydrophilic Jeffamine M1000 (Figure 4). The anti-cancer drugs, CPT or EPI, were encapsulated in the nanocarrier via solvent exchange/precipitation method. CPT-loaded nanocarriers formed 140–200 nm aggregates in PBS (pH 7.4), and CPT solubility showed a 80–100-fold increase. EPI-loaded nanocarriers formed 250 nm aggregates in an aqueous medium, and the multi-drug releases for CPT and EPI release were conducted in PBS (pH 7.4, 37 °C) and monitored for 202 h. An almost linear drug release profile was observed for the initial 8 h; afterwards, a slow drug release was observed for 150 h. The anti-cancer efficacy of these multi-drug loaded nanocarriers was performed on MCF–7 cells, showing higher toxicity than intact CPT and EPI. Meanwhile, almost no toxicity was observed in anti-cancer drug-loaded nanocarriers-treated primary human lung fibroblasts [40].

A pH-sensitive, dual anti-cancer drug delivery carrier was prepared by Örüm et al. They synthesized a PPZ nanosphere that contain curcumin and quercetin through a one-pot drug self-framed precipitation polymerization method. Curcumin and quercetin were used as monomers in the synthesis process. The controlled drug release profile was observed in lysosomal pH (pH 5.5) and blood pH (pH 7.4) at 37 °C. The release rates of quercetin and curcumin from nanospheres were 18.14% and 2.25%, respectively, in pH 7.4 at 7 days. At an acidic pH (pH 5.5), the release rates of quercetin and curcumin were 10.12% and 1.02%, respectively, at 7 days [18].

Mehmood et al. synthesized cross-linked poly(cyclotriphosphazene-*co*-phloretin) (PCTPPT) microspheres. The microsphere was characterized by Fourier transform infrared (FT-IR), and the successful polymerization was confirmed by the newly appeared bands (1009 cm^−1^ and 1132 cm^−1^) in FT-IR. The highly cross-linked structure of the microsphere contributed to their thermal stability. The thermal stability of the microspheres was evaluated by thermogravimetric analysis (TGA), and the results showed the enhanced thermal stability due to the cross-linking. The cumulative drug release profile showed that 41.0% and 32.6% of the drug was released in pH 4.0 and pH 7.4, respectively, after 350 h [19].

Biodegradable DDSs have been attracting significant attention for improving therapeutic efficacy with lesser or negligible side effects. Such a DDS was reported as an acid-sensitive and biodegradable PPZ nano-prodrug using a one-pot cross-linking reaction of 4-hydroxybenzhydrazide-modified doxorubicin (BMD) with HCCP. The phenolic function of BMD showed maximum affinity on HCCP, forming a complex of HCCP-BMD with high drug content of ~86%. The DOX in the prodrug can easily release under acidic conditions, as the hydrazone bonds in BMD and PPZ backbones are highly acid-labile. It was found that the prodrug could exhibit maximum efficacy against HeLa with effective tumor suppression, causing no damage to the healthy tissues [41].

Similarly, a tumor microenvironment (TME)-responsive prodrug system was made of redox/pH multi-responsive and biodegradable PPZ, using a crosslinking of vanillin-modified DOX (VMD, acid-responsive) and 4,4′-dihydroxydiphenyl disulfide (HPS, GSH-responsive) with HCCP. The phenolic functionality of the as-made VMD and HPS have high nucleophilic substitution activity on HCCP under a base catalyst, making the final PPZ nano-prodrug known as HCCP-VMD-HPS (drug content~56.4%). Since the PPZ skeleton and disulfide bonds in HPS and cyclotriphosphazenes are both acid- and GSH-labile, the prodrug can easily be decomposed under TME, releasing the DOX molecules more effectively within the tumor cells under ~10 mM GSH and pH~6.8 than under physiological conditions (pH 7.4 with ~20 μM GSH). The as-made prodrug was able to produce selective toxicity towards cancer cells, with no harm to healthy cells [42].

Similarly, a silane-based, stimuli-sensitive bottlebrush PPZ was made, linked on the surface of mesoporous silica NPs (MSNs). The hybrid polymer, with PEG-like Jeffamine^®^ M–2005 side-arms, could undergo conformational changes under certain pH and temperatures, owing to the amphiphilic and protonatable groups acting as a gatekeeper moiety. Safranin O as control fluorophore or the anti-cancer drug camptothecin (CPT) were loaded within the PPz-coated MSNs. Since the hybrid NPs have characteristic lower critical solution temperature (LCST), the swollen form of PPZ could selectively block the drugs entrapped in the pores. At temperatures above LCST, the PPZ could collapse well, controlling the drug effectively. Moreover, an acidic environment allowed protonation of the polymer skeleton, boosting the pore opening from the surface of the MSNs, enabling the dye release. The as-made NPs were effectively selectively internalized in the A549 tumors when compared to the healthy epithelial BEAS–2B lung cells [43].

Nanosized PPZ-platinum (II) conjugates with varied molecular weights (24,000 to 115,000) have been developed to understand the tumor specificity via the EPR phenomenon and their antitumor activity. It was found that the tumor-homing capability was dependent on the molecular size. These drug conjugates had significant in vivo anti-neoplastic activity on both murine and human cancer cell lines. Most importantly, these PPZ drug conjugates show significant anti-cancer efficacy on YCC–3 (stomach cancer cell line) xenograft model, which is usually hard to treat with traditional therapeutic agents. Such a high anti-cancer activity could be associated with the controlled release of platinum (II) moiety, [GlyGluPt(dach)] (dach = *trans*-(±)-1,2-diaminocyclohexane), from the PPZ backbone via the degradation under aqueous solution [44].

Similarly, cross-linked poly(cyclotriphosphazene-co-luteolin) (PCTPLT) nanospheres were developed using polymerization-induced self-assembly, which was confirmed through FT-IR by observing a specific band at 945 cm^−1^ (P-O-Ar band). Depending on the mole ratio, it was found that not only the shape and size, but also the stability, of NPs could be changed. Moreover, the DOX-loaded NPs had maximum release (~70%) at an acidic pH when compared to a neutral pH (~48%) [45]. This study lacks pre-clinical data using animals.

In another study, poly(cyclotriphosphazene-*co*-hesperetin) microspheres have been synthesized for the 5-fluorouracil delivery. The monomer ratio is crucial here, to determine the stability, size and shape of the micron-sized thermally stable amorphous particles [46]. The major problems associated with drug-loaded PPZ are either related with the lack of pre-clinical data or long-term toxicity analyses.

### 4.2. Bio-Imaging

The intravenous administration of imaging agents is regularly required for patients because of the short half-life of the imaging agents. For long-term imaging, to observe the progress of medical treatment, imaging agent delivery scaffold systems are required; the ideal properties for the delivery scaffold are injectability, biodegradability and sustainability. In these regards, PPZs have been trialed as the imaging agent delivery scaffold. 

A thermosensitive PPZ hydrogel was synthesized for long-term magnetic resonance (MR) imaging. The cobalt ferrite (CoFe_2_O_4_) nanoparticles with hydrophobic surfaces were bound to L-isoleucine ethyl esters on PPZ hydrogel via hydrophobic interactions between the hydrophobic nanoparticles and the -isoleucine ethyl esters on the polymer. The resulting magnetic hydrogel possessed similar properties to the intact PPZ hydrogel, such as viscosity, thermosensitivity, biodegradability, biocompatibility, a reversible solid-to-gel phase transition and injectability. The magnetic hydrogel also showed low cytotoxicity and adequate magnetic properties for use in long-term MR imaging. The magnetic hydrogel was injected into a rat brain, and the long-term imaging capability was successfully tested over 32 days [47].

Overproduction of reactive oxygen species (ROS) is often related to inflammation [48] or cancer [49] and can cause tissue damage. Since the existing ROS imaging techniques have low tissue permeation, it is difficult to perform on the superficial malignancy regions. A specific nanohybrid system based on PPZ and gold nanoparticles (AuNP) has been made for computed tomography (CT) and photoacoustic (PA) imaging applications, as both of these are highly tissue-penetrating imaging tools. The PPZ, being ROS-responsive and biodegradable, could control the PA signal via AuNP disassembly (Figure 5). Such a specific and selective ROS-dependent degradation of NPs could dramatically decrease PA contrast, thereby allowing radiometric ROS detection via comparing the PA/CT signals. Moreover, the ROS imaging was confirmed using an in vitro inflammation model (LPS-stimulated macrophages), and it was found that the ROS-triggered disassembly of the nanoprobe by PA signal. Such novel nanoprobes possess high accuracy and selectivity for ROS imaging, simply via PA to CT signals [50]. Despite the fact that the in vitro results are promising, the lack of in vivo studies on imaging tools and toxicity are major problems associated with these studies.

### 4.3. Gene Delivery

The major challenge to the successful application of gene delivery is gene delivery efficacy and safety of the delivery carrier. The major factor in efficient delivery efficacy is the delivery carrier’s buffering capacity, which affects their endosomal escape. The safety of the delivery carrier is dependent on its biodegradability. Therefore, biodegradable PPZs with controllable side groups are regarded as a suitable gene delivery carrier. 

The first use of PPZs as a gene delivery carrier was tried by Hennink et al. They employed cationic 2-dimethylaminoethanol (DMAE), and the resulting PPZs formed complexes with plasmid DNA (pDNA). At polymer-to-DNA (N/P) ratios above 6, the complexes exhibited positive charge (+25–29 mV) and had a size of 80–90 nm. Intravenous administration of the complexes based on low molecular weight PPZs (Mw 130 kDa) did not show toxicity and resulted in tumor-selective gene expression [51].

The gene delivery efficiency and safety of PPZ-DMAE was further improved by introducing other side groups. Yang et al. introduced imidazole on PPZ-DMAE to make poly(imidazole/DMAEA)phosphazene (PIDP) for enhanced gene delivery efficiency. The PIDP could condense pDNA into complexes of a size around 100 nm and with zeta potential (+25 mV). The transfection efficiency of PIDP/pDNA complex against COS–7, 293T and Hela cells was much higher than that of PPZ-DMAE/pDNA complexes [52]. PEG was conjugated to PPZ-DMAE to improve the safety of the gene delivery carrier. The complexation of PEG-conjugated PPZ-DMAE with pDNA leads to the production of nanometer-sized particles (100~120 nm) and almost neutral particles. The PEG-conjugated PPZ/pDNA complexes showed a 2-fold lower gene delivery efficiency in OVCAR 3 cells when compared to the PPZ-DMAE/pDNA complexes. However, the PEGylated PPZs/pDNA complexes did not induce erythrocyte and excellent shielding of the surface charge of the complexes, while uncoated PPZ-DMAE/pDNA complexes induced aggregation of erythrocytes [53].

PPZ-based hydrogel has been also adopted for gene delivery, since hydrogel can control the release of gene/carrier complexes to overcome the drawbacks associated with the unstable serum level of complexes and their limited therapeutic effect. The thermosensitive and injectable PPZ gels were used for the controlled release of chitosan-PEI/siRNA complexes or PPZ-PEI/siRNA complexes by Kim et al. The released complexes from the hydrogel accumulated in the tumor and showed anti-tumor effects via a single injection in an animal study [54]. 

PPZ-based gene delivery has the potential to be used for immunotherapy, which has recently been attracting attention, along with the development of check point inhibitors. The delivery of gene coding IL–12, which activates cytotoxic T lymphocytes, can contribute to activation of anti-tumor immunity. However, the severe toxicity associated with the systemic delivery of IL–12 limits its local administration; instead, IL–12 gene delivery can be favorable for the prolonged expression and attenuated side effects. Gao et al. prepared complexes for systemic delivery of recombinant murine IL–12 plasmid (pmIL–12) with amphiphilic PPZs containing cationic N,N-diisopropylethylenediamine (DPA) hydrophilic PEG. These complexes showed good stability in serum protein and DNase. The results of an in vivo antitumor study showed that intravenous injection of pmIL–12 complexes achieved significant suppression of tumor growth in a CT–26 colon carcinoma mouse model. After treatment, it was confirmed that immune effector cells, including CD8+ T cells, NK cells and NKT cells, were recruited in the tumor [55].

### 4.4. Protein Delivery

Protein drugs have become an important class of medicines. Recently approved protein drugs have been developed to treat a wide range of diseases, such as cancers, auto-immune diseases and infectious diseases. However, protein drugs are known to have a short half-life and low cell permeability. To overcome these problems, delivery systems have been required to provide protein stabilization and improved intracellular delivery. Andrianov et al. synthesized pH-sensitive PPZ polyelectrolytes containing grafted PEG and either carboxylic acid or tertiary amino groups. These polymers were formulated at similar physiological conditions into nanoassemblies with a size below 100 nm. The complexation of the PPZ-based nanoassemblies with L-asparaginase (L-ASP) resulted in the preservation of enzymatic activity of L-ASP, but improved its thermal stability and resistance against proteolysis [56].

The stability of protein was also overcome by hydrogel-based delivery. A thermosensitive and injectable PPZ hydrogel was synthesized by Sone et al. They induced complexation between negatively charged proteins, such as BSA, gelatin-type B 75 bloom and α-amylase, and hGH and polycations, such as polyethylenimine, poly-L-lysine and poly-L-arginine (PLA), via an electrostatic interaction, and loaded the complexes into PPZ hydrogels. Among polycations, PLA formed the largest complex with negative charged protein and exhibited the slowest release [57]. The biological application of this hydrogel system was further investigated by the same group. The polyelectrolyte complexes between negatively charged hGH and positively charged protamine sulfate (PS) were loaded into an injectable and thermosensitive PPZ hydrogel for a sustained hGH release. The hydrogel system suppressed the initial burst release of hGH, showing an extended release pattern. The pharmacokinetics study in rats revealed that the hydrogel system extended the half-life of hGH 13-fold, and a similar area under the curve (AUC) was observed when compared to hGH solution. The hypophysectomized rats were treated with hGH-loaded PPZ hydrogel by single injection, and an increased growth rate was observed when compared to daily injection of hGH for 7 days [58]. 

This system has also been applied to diabetes, a metabolic disorder requiring life-long treatment after diagnosis. A cationic protamine-conjugated PPZ polymer (ProCP) was developed to form complexes with negatively charged Ex–4. The complexes formed a hydrogel in response to body temperature. The Ex–4/ProCP complexes were slowly released from the hydrogel and Ex–4 was dissociated from dissociation from Ex–4/ProCP complexes, showing a prolonged release pattern in vitro and in vivo. The biological efficacy of the sustained-released Ex–4 was examined with a single subcutaneous injection of the Ex–4/ProCP complex hydrogel in a diabetic mouse model. The blood glucose levels were monitored for 14 days, and fluctuation in the blood glucose level was monitored in the Ex–4 solution group (control). In contrast to the control, Ex–4/ProCP complex hydrogel-treated groups showed extended lowered glucose levels [59]. 

### 4.5. Drug-Free Therapy

pH-responsive drug carriers derived from polymers containing weak base groups have been shown to improve the anti-tumor effect of chemotherapeutics. The common interpretation is that a “proton sponge effect” caused by pH-responsive polymers facilitates endosomal membrane destruction and accelerates cytoplasmic drug release in tumor cells. However, the mechanisms by which pH-responsive weak base polymers disrupt membranes have not been expatiated clearly. Herein, we synthesized a series of pH-responsive amphiphilic polyphosphazenes containing diisopropylamino (DPA) side groups with various contents and investigated the effect of DPA content on the actions of polymers with cell membranes. In a certain pH range, the polymers with elevated DPA content showed enhanced membrane disruptive activity. Electrical interactions between the protonated DPA groups of polymers and the cell lipid bilayer are critical for pH-dependent membrane disruption, which can be competitively prevented by serum proteins. On the other hand, the hydrophobic unprotonated DPA moieties can insert into lipophilic regions of the cell membrane. These synergic actions caused the consequent alteration of bio membrane permeability. More interestingly, it was also found that DPA-rich polymers exhibit higher P-glycoprotein (P-gp) inhibition activity when compared to the polymer containing only low levels of DPA, by efficiently blocking the internal epitope of P-gp. These findings provide strong evidence for the use of pH-responsive amphiphilic PPZs containing DPA side groups as promising drug carriers for intracellular drug delivery applications, especially in the treatment of P-gp-overexpressing, drug-resistant tumors [60].

### 4.6. Phototherapy

Phototherapy can be classified into two categories: photodynamic therapy (PDT), which causes localized chemical damage; and photothermal therapy (PTT), which causes localized thermal damage to the tumor. Both therapies are more precise than other therapies [61,62,63,64,65,66,67]; however, PDT is often hindered by the antioxidant defense system, which depletes the reactive oxygen species (ROS) generated by photosensitizers. 

To overcome this limitation, Lingjie et al. developed a nano-system composed of a photosensitizer (methylene blue, MB), an anti-cancer drug (resveratrol; RV) and a bio-reducible compound (bis-(4-hydroxyphenyl)-disulfide; HPS). The nano-system was constructed by the cross-linking of HCCP with HPS onto the Fe_3_O_4_. The nano-system showed good physiological stability, biocompatibility and a pH/glutathione (GSH)-responsive property and resulting controlled RV and MB release in the tumor microenvironment (TME). The disulfide bond degradation through GSH results in GSH depletion and inhibition of ROS scavenging in TME. This work presents a novel strategy to overcome the ROS scavenging effect in TME and augment efficacy of PDT [68].

A PDT/PTT combination therapy was achieved by Xuan et al. using a PPZ-based nanosphere system. They covalently conjugated photosensitizing porphyrin monomers and HCCPs to form a cross-linked nanosphere. Then, gold nanoparticles were immobilized on the nanosphere surface for photothermal activity. The system was conjugated by PEG for enhanced biocompatibility (Figure 6). The efficient PDT and PTT activity was confirmed by in vitro cell killing studies. The nanospheres exhibited low cytotoxicity against HeLa cells when incubated in dark conditions. However, significant cytotoxicity was observed when HeLa cells were treated with 808 nm laser for PTT and 630 nm LED for PDT. Interestingly, a synergistic PDT/PTT effect was observed under sequential irradiation of 630 nm LED and 808 nm laser (each for 15 min), enabling a maximum therapeutic efficacy through the nanosphere systems via combined PDT/PTT [69]. 

Another approach for PDT/PTT combination therapy was suggested by Tan et al. They developed cyclomatrix polyphosphazene system for dual-modality phototherapy. Briefly, HCCP was conjugated to Zn(II) phthalocyanine (ZnPc) to form dendritic units through the nucleophilic substitution reaction. The nanoparticles were then prepared by incorporation of polyvinylpyrrolidone (PVP). In vitro study results show that the PVP-stabilized nanoparticles undergo the photothermal/photodynamic processes to generate single oxygen and heat, resulting in cancer cell death upon exposure to a single-bandwidth near infrared (NIR) laser (785 nm) [70].

Similarly, multifunctional NPs were made by directly welding superparamagnetic Fe_3_O_4_ NPs and Au shells to highly cross-linked PPZ as “glue” in an easy but effective method (Figure 7). These hybrids can take advantage of MRI diagnosis and strong NIR absorption of Au nanoshell for PTT [71]. Although the vitro results are promising, there is a lack of in vivo confirmation for the same, limiting the applicability in clinical terms.

A major problem associated with PDT is the hypoxia condition in TME, which suppresses the cancer treatment efficacy. To solve this problem, Kai et al. prepared a multimodal yolk-shell nanotheranostic system (PFH–Fe_3_O_4_@PPZ@PDA-PEG-Ce6). The multimodal system was prepared by encapsulation of perfluorohexane (PFH) and Fe_3_O_4_ nanoparticles in PPZ and polydopamine (PDA) shells. Then, after loading chlorine e6 (Ce6) and modifying with PEG, PFH–Fe_3_O_4_@PPZ@PDA-PEG-Ce6 was constructed. The multimodal system facilitated ultrasonic (US), magnetic resonance (MR) and fluorescence imaging-guided PTT and PDT. PFH plays an important role in delivering oxygen and the contrast agent for US imaging. The yolk-shell nanotheranostic showed obvious MRI and US imaging in vitro and in vivo. Moreover, Ce6 generated ROS for PDT upon 660 nm NIR illumination, due to the oxygen delivered by PFH. In sum, a combination of PTT and PDT using a yolk-shell nanotheranostic system yielded a synergistic effect in cancer efficiency [72].

### 4.7. Vaccine

The vaccine has been spotlighted, due to the recent COVID–19 pandemic [73]. Meanwhile, it has been found that the combination of a vaccine and adjuvant is a better option for inducing robust immune response, rather than using a vaccine alone. Although lipid-based adjuvants have been used for vaccines, highly functional polymeric adjuvants are required for generating vaccine combinations tailored to specific pathogens. In this regard, vaccine adjuvant research is another area of application of PPZs [74,75,76,77,78,79,80].

The adjuvant activity of PPZs was identified by the study of poly[di(carboxylatophenoxy)phosphazene] (PCPP) with formalin-inactivated influenza virions. The addition of PCPP enhances the antibody response 10-fold, compared to the levels induced by the vaccine alone [81]. Another type of PPZ, poly[di(sodium carboxylatoethylphenoxy)phosphazene] (PCEP), has an ability to enhance antigen-specific immune responses. In an animal study using BALB/c mice, the group immunized with antigen and PCEP showed significantly enhanced serum antigen-specific antibody titer when compared to the group immunized with antigen alone [82].

The function of PPZs as an adjuvant enabled the promising approach of using PPZs as an injection microneedle for intradermal delivery of a vaccine. Andrianov et al. developed a PPZ-coated microneedle system delivering hepatitis B surface antigen (HBsAg). In studies performed in pigs, the antigen delivered by the PPZ-coated microneedle system showed 10 times higher IgG titers when compared to same formulation administered by intramuscular injection. The synergistic effects between the PPZ-based microneedles and antigen was demonstrated, enabling a minimally invasive transdermal vaccination method [83].

Until recently, various biomaterials have been reported to be equipped with immune-modulatory functions (i.e., either immune-stimulatory or immune-suppressive functions). Due to their structural flexibility, PPZs can be engineered for the development of new immunotherapy strategies. The immune-modulatory PPZs will provide new opportunities for treatment of autoimmune diseases, infectious diseases and cancer. 

### 4.8. Bone Regeneration

PPZ-based biomaterials have also been explored for bone regeneration and bone engineering applications [84]. PPZs are attractive materials in bone regeneration, as they are non-toxic and produce a neutral pH degradation product. For example, hydroxy apatite-PPZ hybrids have been widely explored for bone regeneration applications [85,86]. Deng et al. suggested a PPZ and polyester blend system as an ideal material for bone regeneration [87]. They prepared polyphosphazene poly[(glycine ethyl glycinato)_1_(phenyl phenoxy)_1_phosphazene] (PNGEG/PhPh) and its blends with a polyester. Two polymer blends, Matrix 1 (PNGEG/PhPh:PLGA = 25:75) and Matrix 2 ((PNGEG/PhPh:PLGA = 50:50), were prepared at two different weight ratios of PNGEG/PhPh and PLGA. According to the percentage of mass loss data conducted in phosphate buffered saline at 37 °C over 12 weeks, PLGA was completely degraded at 7 weeks, but the blends showed a slower degradation rate, in the order of Matrix 2 < Matrix 1. During the degradation, acidic degradation products were produced from PLGA and the osteoblast cell number decreased significantly on PLGA matrices; however, enhanced osteoblast cell growth was observed on Matrix 1 and Matrix 2, due to their neutral degradation products. The improved biocompatibility of Matrix 1 and Matrix 2 was also demonstrated in a rat subcutaneous implantation model over 12 weeks. 

For the successful application of PPZ and polyester blend system in bone regeneration, a three-dismensional (3D) scaffold was prepared to mimic the hierarchical architecture and mechanical properties of the natural bone extracellular matrix (ECM) [88]. The fibers in the diameter range of 50–500 nm were produced by the electrospinning method. The process was optimized to produce fibers with suitable mechanical strength and porosity. Using the fibers, 3D scaffolds were prepared, with a central cavity mimicking the bone marrow cavity. The scaffolds showed a two-phase compressive stress/strain behavior, similar to that of natural bone. Subsequent studies were performed to test the mechanical and biological suitability of the scaffold, using primary rat osteoblasts (PRO) cell culture. The PRO cell-seeded scaffold maintained the mechanical properties, while the cell-free scaffold showed the decreased compressive modulus because of the degradation. This means that the cell-mediated matrix production compensated for the loss of the mechanical properties of the scaffold. Similar cellular organization and cell-mediated matrix production to that of natural bone were observed. This study demonstrated the feasibility of a scaffold with a PPZ and polyester blend; however, its bone regeneration capability has not been proved by in vivo study. 

In 2021, Ogueri et al. proved the bone regenerative capability of a 3D scaffold fabricated from a PPZ and PLGA blend using a bone defect animal model [89]. In this study, a 3D scaffold was produced using the blends of poly[(glycine ethylglycinato)_75_(phenylphenoxy)_25_]phosphazene (PNGEGPhPh) and PLGA. Then, in vivo evaluation was performed using a rabbit critical-sized bone defect model, to show the bone tissue regeneration capability of the scaffold. The X-ray images showed that the rabbit model implanted with 3D scaffold fabricated from the blend had higher levels of bone density than the model implanted with PLGA, implying higher rates of new bone formation in the 3D scaffold prepared by the polymer blend. Micro-computed tomography (CT) quantification results indicated that that new bone volume fractions were significantly higher for the 3D scaffold fabricated using blends than of the PLGA after 4 weeks. Histologically, the 3D scaffold-implanted group appeared to show superior bone ingrowths when compared to the PLGA group. In summary, PPZ-containing biomaterials showed excellent potential for use in the application of bone regeneration.

Because of the high ROS in the body injuries, bone defects are found with several inflammatory reactions and therefore regenerative approaches are crucial. Traditionally there have been many conductive polymers used for such applications, however, they are not degradable under in vivo micro-environment, meaning that they are clinically unsafe. Therefore biodegradable polyorganophosphazenes (POPPs) could be a better option as ROS-scavenging agent due to their versatility in both degradability and ease of functionalization. Such a PATGP-type (conductive aniline tetramer (AT) substituted PPZ) having functional side groups of aniline tetramer and glycine ethyl ester was analyzed with PLGA to understand the bone regeneration behavior. Various electrospun nanofibrous samples of PLGA, PLGA/PATGP blend, and PLGA/PATGP core–shell nanofibers were made for in vitro/in vivo analyses, of which PLGA/PATGP core-shell nanofibrous strcture had maximum the ROS scavenging, osteogenic differentiation, neo-bone regeneration character than the other samples. This could be due to the PATGP shell on the PLGA/PATGP nanofiber surface enhancing the bone regeneration properties than the blend nanofibers [90].

Similarly, since amino acid ester PPZs are best known for their non-toxic nature and mechanical stability, making them suitable for in vivo bone engineering application. For load bearable tissue engineering, a PPZ modified with leucine, valine, and phenylalanine ethyl esters have been made. Among them, the PPZ modified phenylalanine ethyl ester had maximum glass transition temperature (41.6 °C), ideal for making a hybrid microspheres with 100 nm sized hydroxyapatite (nHAp). The hybrids were then sintered into 3-D porous scaffold (mean pore diameters: 86–145 µm) with compressive moduli of 46–81 MPa. Additionally these hybrids were able to induce maximum osteoblast cell adhesion, proliferation, and alkaline phosphatase expression, perfect for bone tissue engineering purposes [91].

An enhancement in osteogenic differentiation of bone marrow mesenchymal stromal cells (BMSCs) is one of the crucial steps involved in achieving bone regeneration. To do so, a biodegradable simvastatin-bearing PPZ prodrug NPs was made. Additional groups such as tryptophan ethyl ester (photoluminescent) and hydrolyzable glycine ethyl ester were substituted on the PPZ backbone. The resultant polymer, poly(simvastatin-co-ethyl tryptophanato-co-ethyl glycinato)phosphazene (PTGP-SIM), had not only photoluminescence and degradability but also it could enable a sustained release. The intrinsic photo-luminescence of this PTGP-SIM NPs could enable the cellular trafficking in BMSCs effectively. Such a strategic approach could enhance the osteogenic differentiation majorly even with low local concentration [92].

### 4.9. Suface Modification with PPZs

PzF (poly[bis(trifluoroethoxy)phosphazene]) polymer is a soft rubber-like inorganic, high molecular weight fluoropolymer that possesses a backbone of alternating nitrogen and phosphorus atoms and trifluoroethoxy side groups [93]. Early work by Welle et al. showed PzF coating results in hydrophobic surface properties, causing high adsorption of serum albumin, but low adsorption of fibronectin and fibrinogen [94]. PzF polymer is stable in contact with blood and retains its mechanical properties over at least 24 months [90]. In addition, the trifluoroethanol component is known to have anti-inflammatory properties and to stabilize proteins in solution against thermal denaturation [95].

Major advances have been made in coronary artery stent technology over the last decades. Drug-eluting stents reduced in-stent restenosis and have shown better outcomes compared with bare metal stents, yet some limitations still exist to their use. Because they delay healing of the vessel wall, longer dual antiplatelet therapy is mandatory to mitigate against stent thrombosis and this limitation is most concerning in subjects at high risk for bleeding. The COBRA PzF nanocoated coronary stent has been associated with accelerated endothelialization relative to drug-eluting stents, reduced inflammation and thromboresistance in preclinical studies, suggesting more flexible dual antiplatelet therapy requirement with potential benefits especially in those at high bleeding risk. Here, we discuss the significance of COBRA PzF in light of recent experimental and clinical studies [96].

In an early study, it was found that PPZ nanocoat could mitigate thrombogenicity, in-stents, and inflammatory response in porcine renal and iliac artery stents [97].

### 4.10. Dental Applications

Apart from the dug delivery applications, PPZs have been used for various other purposes including dental and tissue engineering purposes. For example, elastomers based on the PPZ skeletal paltforms have been made for such purposes. Alkoxy groups (2,2,2-trifluoroethoxy (CF_3_CH_2_O−) and nonfluorinated alkoxy groups such as CH_3_CH_2_O–, CH_3_CH_2_CH_2_O–, or CH_3_CH_2_CH_2_CH_2_O– have been introduced in to PPZ to aid an access to elastomers with very low glass transition temperatures (Tg)~−60 to −90 °C. It was found that the changes in the contents of various alkoxy and trifluoroethoxy side groups could significantly alter the physical properties as indicated by the stress–strain, differential scanning calorimetry, and thermogravimetric analysis techniques. Such PPZ elastomers could be further beneficial for cardio ad well as dentistry targeted applications [98].

Even though, photo-activable polymer-based dental composites have more preference in restoring carious teeth, the longevity of these composites could be challenging due to the weak bonding of composite with tooth surface. To overcome this issue, the tooth surface was modified with novel fluorinated PPZs, hence improving the interface between composite and tooth surface. It was confirmed that the fluorinated PPZs have good binding isotherms with two major components of teeth collagen (CLG) and hydroxyapatite. Additionally, the modified PPZs have hemocompatibility, no demineralization and hydrophobicity and could restrict the CLG disintegration under acidic condition. This novel system could be a useful technology for making novel dental restorative agents [99].

Another reported restoration agent was based on cyclotriphosphazenes containing 4-allyl–2-methoxyphenoxy and β-carboxyethenylphenoxy moieties with acrylate dental restorative compositions. The successful synthesis was confirmed by ^1^H and ^13^C NMR and MALDI-TOF analyses. The exact optimization for combining the modifier with the initial dental components such as bisphenol A glycidyl methacrylate (bis-GMA) and tri- ethylene glycol dimethacrylic ester (TGM–3) and were unlocked by DSC (differential scanning calorimetry) tool. The developed hybrids could increase the dental tissue adhesion along with curing the depth and the lowered water sorption and solubility. It was also found that the composites could have improved mechanical properties with respect to the concentration of modifier in the hybrids [100].

An iodine-containing cyclophosphazenes have been developed as radiopacifiers in making dental restorations. The hypothesis was cyclophosphazenes containing iodine and acrylate groups could have radiopacity along with good mechanical stability. The nucleophilic substitution of HCCP with hydroxyethyl methacrylate (HEMA) and 4-iodoaniline formed cyclophosphazene. The iodine-cyclophosphazenes were dissolvable in Bis-GMA/triethylene glycol dimethacrylate (TEGDMA) resin (10 or 15%wt. of the resin). Prior to characterizations, developed resins were treated for photo-curing, and obtained composites could achieve X-ray blocking property. Since HEMA-co-iodoaniline substituted cyclotriphosphazenes additive could not produce impressive mechanical properties for the composites as the cyclotriphosphazenes might well blended with the resins. On the other hand, the PPZ rings between resin backbones could restrict the polymerization shrinkage. These experimental results demonstrating that such iodinated cyclotriphosphazenes could be better additives than the conventional ones for developing radiopaque dental resins [101].

Similarly, when hexa-p-carboxyphenoxycyclotriphosphazene reacted with glycidyl methacrylate, an oligophosphazenes containing carboxy and polymerizable methacrylate functionalities were made. When these oligomers (upto 10 wt %) incorporated into methacrylate dental fillings, their tooth tissue adhesion as well as metal adhesion could be increased (5–7×) having compression strength by 20–30% [102]. A similar dental filler composite was made as dental modifiers using hexa-p-hydroxymethylphenoxycyclotriphosphazene and maleic anhydride. The as made oligophosphazenes containing carboxyl groups and co-polymerizable double bonds were used as modifiers of methacrylate dental composites had high adhesion to hard dental tissues and metals as well [103].

### 4.11. Applications in Medical Devices

Dental liners have been used in dentistry to reshape prostheses surfaces contacting with tissues of the oral cavity. The loss of adhesiveness or hardness due to the changes in the physical properties of dental liner can cause problems such as loss of durability and microbial growth [104]. The most widely used liners has been made of silicone polymers and such liners have been reported to lose adhesion and support fungal growth [105]. A PPZ-based dental liner was expected to solve the problems associated with current silicone-based dental liners. In this regard, a PPZ-based resilient liner, Novus (Hygenic Corp, Dayton, OH, USA) has been introduced and clinically evaluated. Novus was made of fluoroalkoxy-substitued PPZ and Di-trimethacylates was added to this formulation to improve the physical properties including hardness, bond strength, tensile strength, bond strength, and water-absorption. Clinical evaluation results of Novus-lined dentures indicated decreased numbers of microbial colonization in comparison with the silicone-based Molloplast-B (Regncri GmbH, Köln, Germany). Historically, the denture liner has been recorded as the first use of PPZs in medical device [106].

### 4.12. Other Applications

Wound dressings with excellent adhesiveness, antibacterial, self-healing, hemostasis properties, and therapeutic effects have great potential in treating an acute trauma. Until now, several mussel-inspired catechol-based wet adhesives have been used despite they are easily oxidized, limiting their applicability. To solve such limitations, a PPZ and non-catechol based injectable antibacterial first aid hydrogel bandage was made. Influenced by barnacle cement proteins, a series of dynamic phenylborate ester based adhesive hydrogels were made through the combination of cation-π structure modified PPZ with polyvinyl alcohol (PVA). The as made wound dressings have good antibacterial property (4 h antibacterial rate 99.6 ± 0.2%), good mechanical stability with good hemostatic maintenance. The hydrogels could strongly attach on the tissue surfaces via cation-π and π-π interactions along with strong hydrogen bonding (adhesion strength = 45 kPa). The in vivo results using the hydrogels showed a shortened bleeding time (reduced by 88%) enhancing the healing process [107].

PPZs have been used as scaffolds for constructing ligands-coordinated-metallic drugs. To exemplify, a coordination chemistry of the (amino)cyclotriphosphazene ligand, [N_3_P_3_(NHCy)_6_], towards gold(I) complexes was studied. Neutral complexes, [N_3_P_3_(NHCy)_6_{AuX}_n_] (X = Cl or C_6_F_5_; n = 1 or 2) (**1–4**), cationic complexes, [N_3_P_3_(NHCy)_6_{Au(PR_3_)}_n_](NO_3_)_n_ (PR_3_ = PPh_3_, PPh_2_Me, TPA; n = 1, 2 or 3) (**6–12**) [TPA = 1,3,5-triaza–7-phosphaadamantane] and a heterometallic compound [N_3_P_3_(NHCy)_6_{Au(PPh_3_)}_2_{Ag(PPh_3_)}](NO_3_)_3_ (**13**) have been made and sample **7**, showed coordination of gold atoms to the nitrogens of the PPZ ring. Sample (**1, 4, 6–13**) have been tested for anti-cancer activity against human cell lines such as MCF7 (breast adenocarcinoma) and HepG2 (hepatocellular carcinoma).To understand the antimicrobial potency of the mentioned compounds, several bacterial species (Gram-positive, Gram-negative, and Mycobacteria) were used. It was found that both IC_50_ and minimum inhibitory concentration (MIC) were lowest for the gold or silver derivatives against the cell lines and especially for Gram-positive (*S. aureus*) strain and the mycobacteria. The structure–activity relation studies revealed the influence of ancillary ligands and the number and type of metal atoms (silver or gold). Samples **4** and **8** had highest potency tumor specific toxicity compared to other samples. On the other hand, anti-bacterial activity of sample **13** (with both gold and silver atoms) was excellent against both Gram-positive and Gram-negative strains (nanomolar range). Such modified PPZ could be extended for various medical fields where anti-biotic behavior is highly preferred [108].

Similar study was reported by Yilmaz et al. (2022), where, monodispersed silica NPs were made by Stöber method. The surface functionalized NPs had excellent drug release and antibacterial properties. Initially, –NH_2_ functionalities were made silica NP surface using (3-aminopropyl)triethoxysilane (APTES). Nextk, HCCP molecules were linked on the silica NP surfaces via –NH_2_ groups on the silica surface. Finally, the chloride groups in the HCCP structure were modified with trimethoprim (TMP) to induce anti-bacterial activity against Escherichia coli, Bacillus subtilis, and Staphylococcus aureus bacteria. The model drug rhodamine 6G was released out by the Korsmeyer–Peppas low power model and non-Fickian release patterns. Both loaded and vehicle NPs were able to induce good antibacterial activity on aforementioned bacterial strains [109].

## 5. Challenges and Perspectives

In most of the reported cases, it is clear that the pre-clinical studies are missing therefore utmost care should be provided in the future. In addition, especially when hybridizing or conjugating with the bio-active molecules, long term toxicity should be clearly understood.

In many of the reported drug delivery, bone-regeneration, dental liners, neither animal studies nor clinical data are given, making it difficult to draw a conclusion on the clinical reliability of PPZ and their hybrids. Attention should also be given on utilizing PPZ towards phototherapy and further extending its applications towards other drug delivery applications (vaccine and anti-biotic applications).

## 6. Conclusions

The research progress of PPZs in drug delivery has been reviewed, focusing on the recent developments. It was found that the most striking property associated with PPZ is their structural diversity, which can control the new co-polymers synthesis. While hydrophilic side chain substituted PPZ has good water solubility, the hydrophobic one has poor solubility. Both hydrophilic/hydrophobic group substituted PPZ form self-assembled micelles which are passively targeted for cancer tissues. Moreover, the PPZ-drug conjugates are able to control the initial burst release in many reported studies. Similarly, the bioavailability of proteins and genes have been improved after encapsulating with PPZ modified with counter-ion groups. Interesting properties such as thermo-sensitivity can be given to PPZ upon side chain substitution with thermo-responsive groups, which can show a sol–gel transition, and can be used as hydrogels with enhanced biodegradability for adjuvant therapy. Drugs in PPZ-hydrogels can easily home to tumor site via degradation of its backbone.

In addition to the aforementioned various drug delivery applications, there are several approaches for using PPZs (linear and cyclo ones) in bone regeneration, dental applications, and phototherapy. Although PPZ have multifaceted properties, the strict and specific synthetic conditions (anoxic reaction) make it limited for further development of PPZ hybrids.

Regulatory consideration have also been a concern on PPZ-based drug delivery system development. Although PPZ-based drug delivery system do not have a long history in the medical field, their good biocompatibility data supported by several studies suggests that PPZ-based drug delivery system could be translated to clinical field.

However, it is expected that accumulated safety data will encourage to consider PPZs as good materials for the future design of drug delivery system. In addition, the use of PPZs is expected in the personalized medicine, which has recently been in the spotlight. The versatility in design, good biocompatibility and controllable degradation characteristics of PPZ can be designed to obtain highly specific personalized medicines in the very near future.

## Figures and Tables

**Figure 1 ijms-23-15993-f001:**
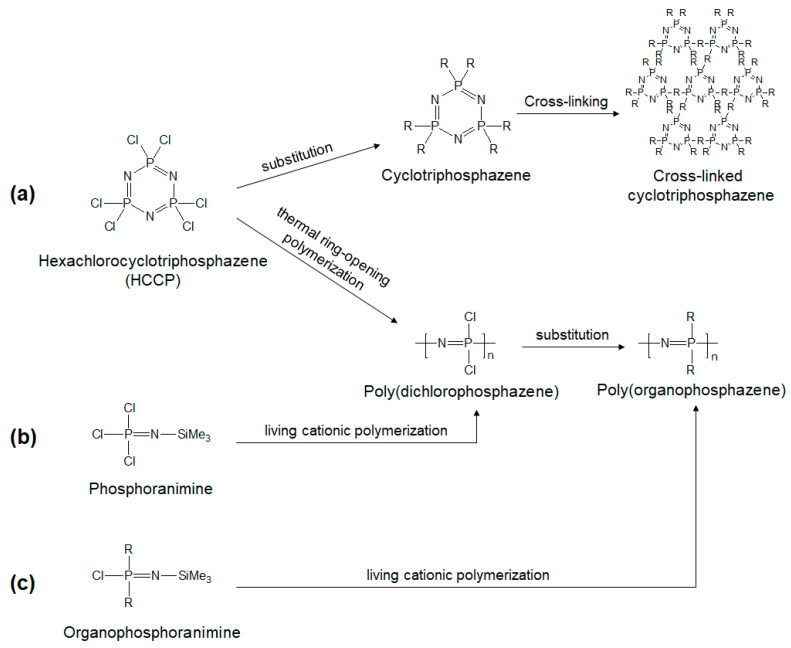
The synthetic routes of PPZs: ring-opening polymerization, living cationic polymerization and substitution reaction are the major techniques adopted. Synthetic route of PPZs from (**a**) HCCP, (**b**) phosphoranimine and (**c**) organophosphoranimine.

**Figure 2 ijms-23-15993-f002:**
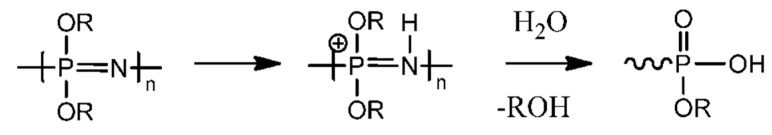
Acid catalyzed degradation mechanism. Figure 2 is adapted and reused from [24] with permission from MDPI, 2013, under Creative Commons Attribution license.

**Figure 3 ijms-23-15993-f003:**
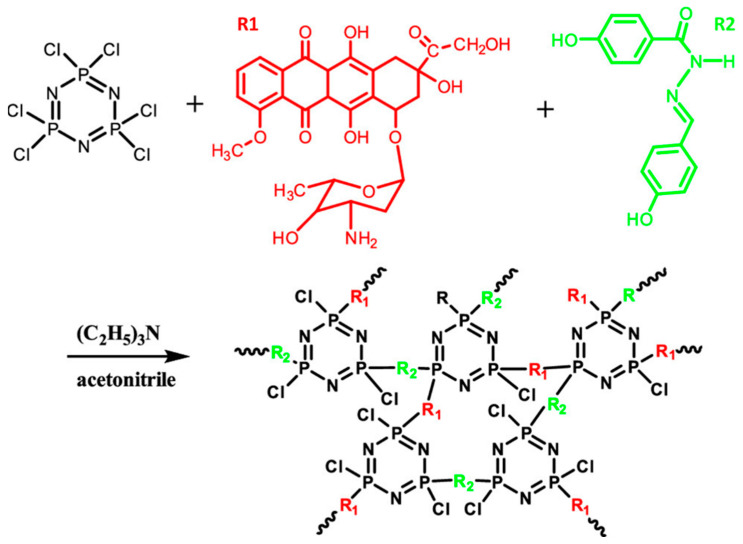
Synthesis route for cross-linked PPZ nano-drugs. Adapted and reused with permission from [38], 2020, American Chemical Society.

**Figure 4 ijms-23-15993-f004:**
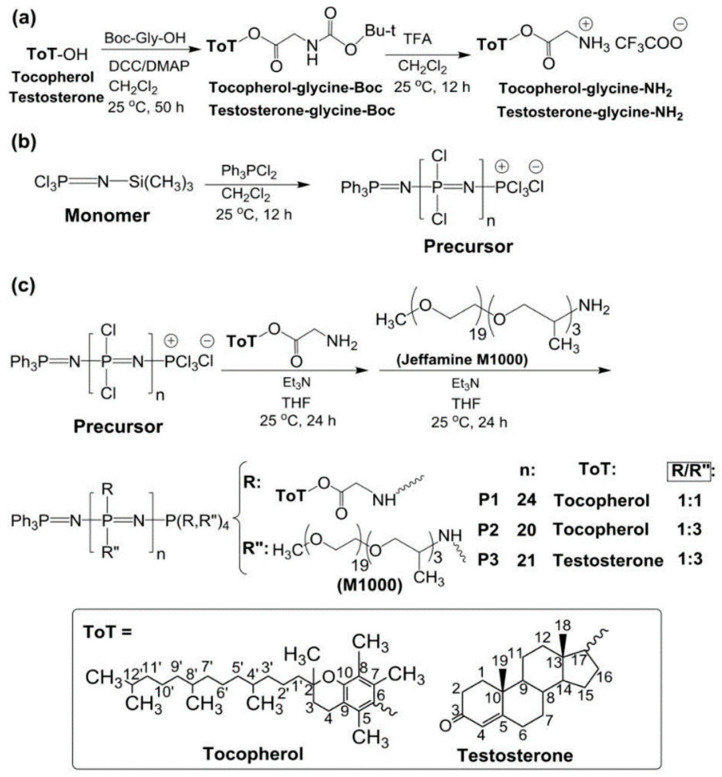
(**a**) Synthesis of tocopherol and testosterone glycinates (ToT-glycine-NH_2_). (**b**) Polymerization of trichlorophosphoranimine to obtain poly(dichloro)phosphazene. (**c**) Post-polymerization process of chlorine atoms with ToT-glycine-NH_2_ and Jeffamine M1000. Copyright 2022 Licensee MDPI, Basel, Switzerland, under the terms and conditions of the Creative Commons Attribution (CC BY) license (https://creativecommons.org/licenses/by/4.0/, accessed on 18 October 2022) [40].

**Figure 5 ijms-23-15993-f005:**
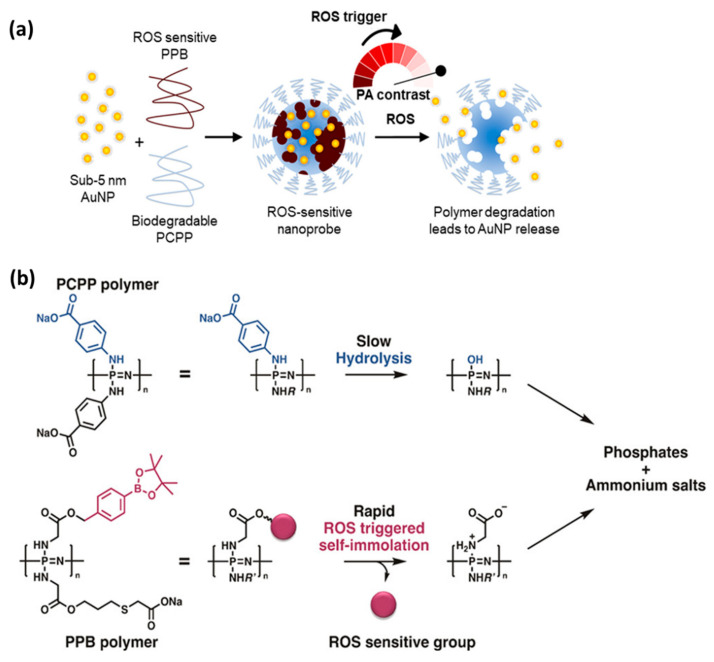
(**a**) Assembly of AuNP in nanogels that selectively degrade under ROS exposure and release free AuNP for imaging of ROS with PA; (**b**) Degradation pathways of PCPP (Upper) and the ROS-sensitive PPB (lower) yielding neutral pH and nontoxic byproducts. Adapted and reused from [50], American Chemical Society, 2019.

**Figure 6 ijms-23-15993-f006:**
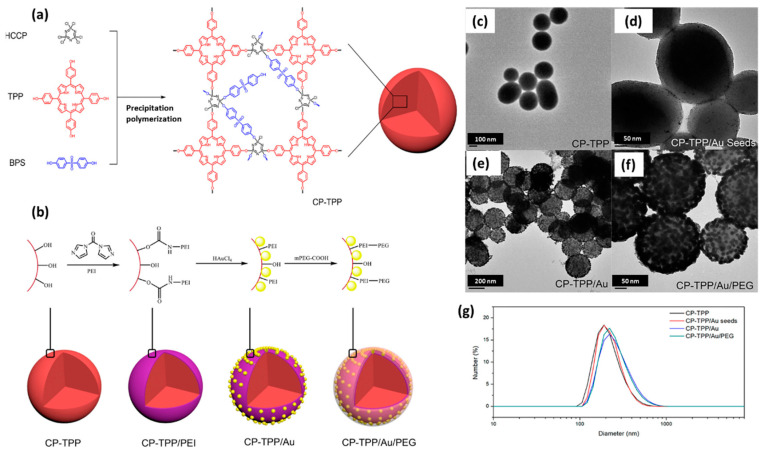
(**a**) Synthetic route and proposed chemical structure of cross-linked CP-TPP nanospheres; (**b**) Synthetic procedure for preparing CP-TPP/Au/PEG nanospheres and (**c**–**f**) TEM images and (**g**) size distribution of CP-TPP, CP-TPP/Au seeds, CP-TPP/Au, and CP-TPP/Au/PEG nanospheres based on DLS. Adapated and reused with permission from [69], American Chemical Society, 2018.

**Figure 7 ijms-23-15993-f007:**
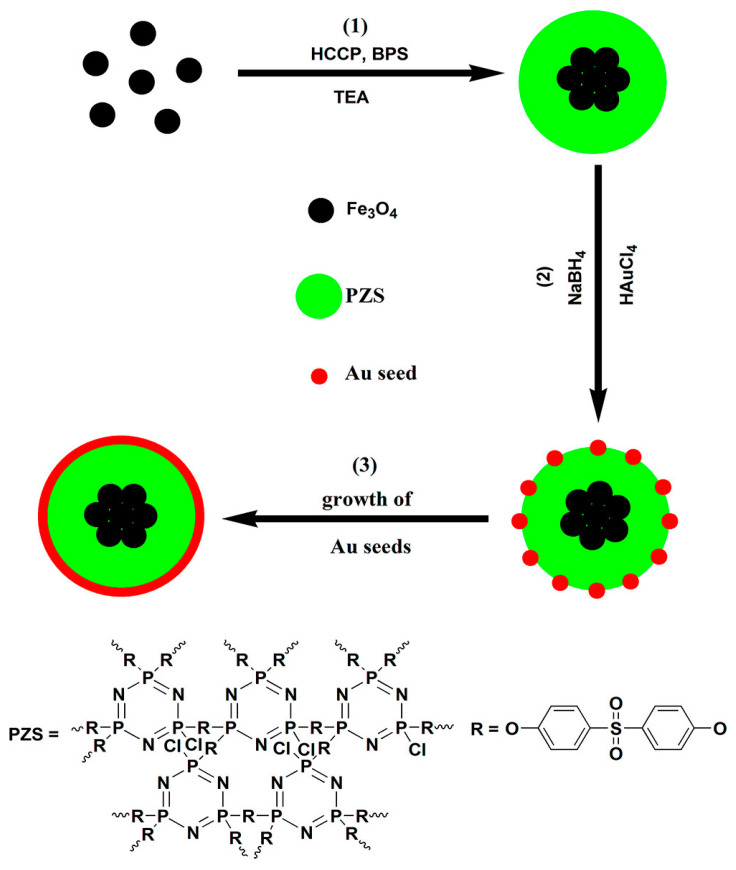
Preparation procedure of Fe3O4@PZS@Au Shells [71].

**Table 1 ijms-23-15993-t001:** Recently reported various polyphosphazene-based various drug delivery systems.

Drug Delivery Cargo(s)	Key Synthetic Mechanism/Conjugation	Key Findings	Reference
DOX	Thermal ring-opening polymerization	Amphiphilic graft PPZs for DOX delivery	[15]
Fluorescein and quercetin	One-pot polycondensation of octachlorocyclotetraphosphazene, fluorescein and quercetin.	379 nm sized NPs with enhanced stability under aqueous and organic solvents with improved fluorescence	[16]
Curcumin and quercetin	self-assembly polycondensation reaction	230–600 nm sized NPs with dual delivery properties	[17]
DOX with mitoxantrone (Mit), resveratrol (RES), curcumin (Cur), and porphyrin (TPP)	Drug self-framed delivery system (DSFDS)	<200 nm spherical particles	[18]
Camptothecin (CPT)	Cross-linked poly(cyclotriphosphazene-*co*-phloretin) (PCTPPT) microspheres	Controlled drug delivery systems for CPT	[19]
Ovalbumin	Self-assembly method	Phospholipid-like polyphosphazene amphiphilic NPs for oral delivery of ovalbumin antigen for immunotherapy	[20]

## Data Availability

Not applicable.

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
