# Peer review of "Polyphosphazene-Based Biomaterials for Biomedical Applications"

_ijms, 2022, doi:10.3390/ijms232415993_

Round 1

Reviewer 1 Report

The manuscript cannot be published in the International Journal of Molecular Sciences without significant revision.

First, the authors need to add critical comments to the reviewed articles. At the moment, the review looks just like a description of what was done in the articles reviewed by the authors without the necessary analysis of them.

Secondly, the amount of reviewed literature is insufficient. The small number of cited literature allows us to conclude that the topic of the review is not relevant. It should be borne in mind that you are submitting your manuscript to a fairly high-ranking journal. Therefore, it is necessary to add another 40-50 scientific references to the review. Perhaps the authors need to expand the scope of the review and go beyond linear polyphosphazenes. It is worth considering polymers containing cyclophosphazenes, which are also widely used. Also, one should not be limited to the use of polyphosphazenes for drug delivery. The use of phosphazenes in medicine and dentistry could also be considered. If this is not done, the review seems extremely poor for the International Journal of Molecular Sciences.

There are also comments on the design of the article. Figure 1 needs to be corrected. The correct spelling is hexachlorocyclotriphosphazene (line 42).

Author Response

all the revisions have been marked in the red colour in the revised manuscript

Reviewer 2 Report

The manuscript reviews the recent literature on the use of polyphosphazenes as matrixes for drug delivery systems. The authors start with the historical background of such polymers, citing the original references, and move forward to recent reports on their application in drug delivery.

However, before being considered for publication, the authors must address the following issues:

-          The English language needs to be thoroughly revised, as many sentences are beyond understanding.

-          The authors should make sure that the information presented is enough to give the reader a full overview of the cited work; for example, in lines 41-45 the authors describe the thermal ring opening polymerization of HCCP with two different outcomes without providing any explanation.

-          The authors may summarize, in the form of a table like Table 1, the information presented in the text, but the presentation of information in a table does not replace its description in the text. In Table 1, only the work of reference [9] is described in the text; that of references [10] to [14] is not.

-          The statement: “The biocompatibility is strongly supported by the fact that a coronary stents, Cobra 125 PzF stents (CeloNova Biosciences Inc., US) coated with PPZs has already been approved 126 by FDA” in lines 125-127 needs a reference.

-          Some acronyms are not defined prior to their use.

-          The authors must check if all the information is correct; for example, the size of PPZ-DTX micelles is not 41.38 nm (line 141), but 41.8 nm.

-          In Figure 1, the nature of R groups should be stated; a “n” is missing after the brackets in the polymers structures; the end groups of the polymers are also missing.

-          In Figure 3, the structures of DOX and HBHBH should be labelled R1 and R2, respectively.

-          The captions of Figures 2 and 4 need to be checked; the resolution of Figure 5 is too low.

Author Response

All the revised parts have been marked in red colour 

Round 2

Reviewer 1 Report

The authors have significantly improved their manuscript, but I have a few small but very significant comments that should be eliminated.

Abstract: Since there is no content in the review, it is very difficult for the reader to understand what is being discussed in the review. Authors should clearly state in the abstract which topics the review is devoted to and what they focus on. This is very important to get more audience and review cited.

Dentistry section: In dentistry, phosphazenes find much more use, for example as adhesion promoters, this should be mentioned in the review (https://doi.org/10.3390/polym12051176; https://doi.org/10.1134/S1070427215050225; https: //doi.org/10.1134/S156009041306002X)

The conclusion does not reflect the sections added by the author. The conclusion should be expanded.

Author Response

We thank reviewer 1 for the valuable comments and suggestions. All the revised portions have been done in BLUE colour for better understanding.

Reviewer 2 Report

The reviewer understands that the authors are not native English speakers. However, the level of the English language remains too low for a good understanding of the manuscript content by the readers. The improvements made to the English language in the revised version are mininal.

As some examples:

- "PPZs are hydrolytic unstable"

- "chloride atom" (chloride is an ion, the atom is chlorine)

- "enriching the synthetic routes"

- "it is further possible for chemical reaction"

- "PPZs has a hydrolytic labile properties"

- "The attempts has been made"

- "The biodegradability of PPZs is beneficial property"

- "This degradation products is advantage"

- "In the studies, osteoblast cells on PPZ polymer surfaces normal characteristic of osteoblast phenotype, differentiation, maturation, and mineralization."

- "The PPZ-DTX formed nanometer size of stable micelles"

- "Another problem associated with DOX is a resistance which has severely hindered its clinical application"

- "Against human breast cancer cell line (MCF-7), anticancer drug-loaded nanocarriers exhibited similar or superior toxicity when compared to raw CPT and EPI."

- "Post-polymerization functionalization of chlorine atoms" (one does not functionalize atoms, chlorine or other)

- "the same properties with the original PPZ hydrogel"

etc.

The reviewer does not agree with the answer given to the 3rd question, as it should be all the way around: all the examples referred by the authors should be thoroughly described in the text and the most relevant ones highlighted in a Table.

Author Response

We thank reviewer 2 for the valuable comments and suggestions. All the revised portions have been done in RED colour for better understanding.

Round 3

Reviewer 1 Report

The authors took into account all comments.